# "Best-of-Many-Samples" Distribution Matching

## Abstract

Generative Adversarial Networks (GANs) can achieve state-of-the-art sample quality in generative modelling tasks but suffer from the mode collapse problem. Variational Autoencoders (VAE) on the other hand explicitly maximize a reconstruction-based data log-likelihood forcing it to cover all modes, but suffer from poorer sample quality. Recent works have proposed hybrid VAE-GAN frameworks which integrate a GAN-based synthetic likelihood to the VAE objective to address both the mode collapse and sample quality issues, with limited success. This is because the VAE objective forces a trade-off between the data log-likelihood and divergence to the latent prior. The synthetic likelihood ratio term also shows instability during training. We propose a novel objective with a "Best-of-Many-Samples" reconstruction cost and a stable direct estimate of the synthetic likelihood. This enables our hybrid VAE-GAN framework to achieve high data log-likelihood and low divergence to the latent prior at the same time and shows significant improvement over both hybrid VAE-GANS and plain GANs in mode coverage and quality.

## 1 Introduction

Generative Adversarial Networks (GANs) (Goodfellow et al., 2014) have achieved state-of-the-art sample quality in generative modeling tasks. However, GANs do not explicitly estimate the data likelihood. Instead, it aims to "fool" an adversary, so that the adversary is unable to distinguish between samples from the true distribution and the generated samples. This leads to the generation of high quality samples (Adler & Lunz, 2018; Brock et al., 2019). However, there is no incentive to cover the whole data distribution. Entire modes of the true data distribution can be missed – commonly referred to as the mode collapse problem.

In contrast, the Variational Auto-Encoders (VAEs) (Kingma & Welling, 2014) explicitly maximize data likelihood and can be forced to cover all modes (Bozkurt et al., 2018; Shu et al., 2018). VAEs enable sampling by constraining the latent space to a unit Gaussian and sampling through the latent space. However, VAEs maximize a data likelihood estimate based on the $L_1/L_2$ reconstruction cost which leads to lower overall sample quality – blurriness in case of image distributions. Therefore, there has been a spur of recent work (Donahue et al., 2017; Larsen et al., 2016; Rosca et al., 2019) which aims integrate GANs in a VAE framework to improve VAE generation quality while covering all the modes. Notably in Rosca et al. (2019), GANs are integrated in a VAE framework by augmenting the $L_1/L_2$ data likelihood term in the VAE objective with a GAN discriminator based synthetic likelihood ratio term.

However, Rosca et al. (2019) reports that in case of hybrid VAE-GANs, the latent space does not usually match the Gaussian prior. This is because, the reconstruction log-likelihood in the VAE objective is at odds with the divergence to the latent prior (Tabor et al., 2018) (also in case of alternatives proposed by Makhzani et al. (2016); Arjovsky et al. (2017)). This problem is further exacerbated with the addition of the synthetic likelihood term in the hybrid VAE-GAN objective – it is necessary for sample quality but it introduces additional constraints on the encoder/decoder. This leads to the degradation in the quality and diversity of samples. Moreover, the synthetic likelihood ratio term is unstable during training – as it is the ratio of outputs of a classifier, any instability in the output of the classifier is magnified. We directly estimate the ratio using a network with a controlled Lipschitz constant, which leads to significantly improved stability. Our contributions

in detail are, 1. We propose a novel objective for training hybrid VAE-GAN frameworks, which relaxes the constraints on the encoder by giving the encoder multiple chances to draw samples with high likelihood enabling it to generate realistic images while covering all modes of the data distribution, 2. Our novel objective directly estimates the synthetic likelihood term with a controlled Lipschitz constant for stability, 3. Finally, we demonstrate significant improvement over prior hybrid VAE-GANs and plain GANs on highly muti-modal synthetic data, CIFAR-10 and CelebA.

## 2 RELATED WORK

**Generative Autoencoders.** VAEs (Kingma & Welling, 2014) allow for generation by maintaining a Gaussian latent space. In Kingma & Welling (2014), the Gaussian constraint in applied point-wise and latent representation of each point is forced towards zero. Adversarial Auto-encoders (AAE) (Makhzani et al., 2016) and Wasserstein Auto-encoders (WAE) (Arjovsky et al., 2017) tackle this problem by an approximate estimate of the divergence which only requires the latent space to be Gaussian as a whole. But, the Gaussian constraint in (Arjovsky et al., 2017; Kingma & Welling, 2014; Makhzani et al., 2016; Mahajan et al., 2019) is still at odds with the data log-likelihood. In this work, we enable the encoder to maintain both the latent representation constraint and high data log-likelihood using a novel objective. Furthermore, we integrate a GAN-based synthetic likelihood term to the objective to enhance the sharpness of generated images.

**Mode Collapse in Classical GANs.** The classic GAN formulation (Goodfellow et al., 2014; Radford et al., 2016) has several shortcomings – importantly mode collapse. Denoising Feature Matching (Warde-Farley & Bengio, 2017) deals with the mode collapse by regularizing the discriminator using an auto-encoder. MDGAN (Che et al., 2017) uses two separate discriminators and regularizes using a auto-encoder. In EBGAN (Zhao et al., 2017a), the discriminator is interpreted as an energy functional and is also cast in an auto-encoder framework, leading to improvements in semi-supervised learning tasks. BEGAN (Berthelot et al., 2017) proposes a Wasserstein distance based objective to train such GANs with auto-encoder based discriminators. The proposed approach leads to smoother convergence. InfoGAN (Chen et al., 2016) maximizes the mutual information between a small subset of latent variables and observations in a Information Theoretic framework. This leads to disentangled and more interpretable latent representations. PacGAN (Lin et al., 2018) proposes to deal with the mode collapse problem by using the discriminator to distinguish between product distributions. D2GAN (Nguyen et al., 2017) proposes to use two discriminators – one for the forward KL divergence between the true and generated distributions and one for the reverse. BourGAN (Xiao et al., 2018) proposes to learn the distribution of the latent space (instead of assuming Gaussian) which reflects the distribution of the data. In (Srivastava et al., 2017), a inverse mapping from from latent to data space is learned and the generator is penalized based on the inverted distribution to cover all modes. Ravuri et al. (2018) proposes a moment matching paradigm different from VAEs or GANs. However, as the presented moment matching network involves an order of magnitude more parameters compared to VAEs or GANs, we do not consider them here. As we propose a hybrid VAE-GAN framework these techniques can be applied on top to potentially improve results. However, in hybrid VAE-GANs the reconstruction loss already incentivizes the coverage of all modes.

**Wasserstein Loss based Formulations.** Arjovsky et al. (2017); Gulrajani et al. (2017) proposes GANs which minimize the Wasserstein distance between true and generated distributions. Miyato et al. (2018) demonstrates improved results by applying Spectral Normalization on the weights. In Tran et al. (2018), distance constraints are applied on top. In Adler & Lunz (2018) WGANs were extended to Banach Spaces to emphasize edges or large scale behavior. Orthogonally, Karras et al. (2018) focus on progressively learning to use more complex model architectures to improve performance. We use the regularization techniques developed for WGANs to improve stability of our hybrid VAE-GAN framework. Brock et al. (2019) shows very high quality generations at high resolutions but these are class conditional. However, diverse class conditional generation is considerably easier as intra-class variability is generally much lower than inter-class variability. Here, we focus on the more complex unconditional image generation task.

**Hybrid VAE-GANs.** In Larsen et al. (2016) a VAE-GAN hybrid is proposed with discriminator feature matching – the VAE decoder is trained to match discriminator features instead of a $L_1/L_2$ reconstruction loss. ALI (Dumoulin et al., 2016) proposes to instead match the encoder and decoder joint distributions – with limited success on diverse datasets. BiGAN (Donahue et al., 2017), builds

upon ALI to learn inverse mappings from the data to the latent space and demonstrate effectiveness on various discriminative tasks. Rosca et al. (2019) extends standard VAEs by replacing the log-likelihood term with a hybrid version based on synthetic likelihoods. The KL-divergence constraint to the prior is also recast to a synthetic likelihood form, which can be enforced by a discriminator (as in Makhzani et al. (2016); Tolstikhin et al. (2018)). The second improvement is crucial in generating realistic images at par with classic/Wasserstein GANs. We further improve upon Rosca et al. (2019) by allowing the encoder multiple chances to draw desired samples and enforcing stability – enabling it to maintain low divergence to the prior while generating realistic images.

## 3 Novel Objective for Hybrid VAE-GANs

We begin with a brief overview of hybrid VAE-GANs followed by details of our novel objective.

**Overview.** Hybrid VAE-GANs (Figure 1) are generative models for data distributions $x \sim p(x)$ that transform a latent distribution $z \sim p(z)$ to a learned distribution $\hat{x} \sim p_\theta(x)$ approximating $p(x)$. The GAN ($G_\theta$, $D_I$ alone can generate realistic samples, but has trouble covering all modes. The VAE ($R_\phi$, $G_\theta$, $D_L$) can cover all modes of the distribution, but generates lower quality samples overall. VAE-GANs leverage the strengths of both VAEs and GANs to generate high quality samples while capturing all modes. We begin with a discussion of the prior hybrid VAE-GAN objectives (Rosca et al., 2019) and its shortcomings, followed by our novel "Best-of-Many-Samples" objective with a novel reconstruction term and regularized stable direct estimate of the synthetic likelihood.

### 3.1 Shortcomings of Hybrid VAE-GAN Objectives

Hybrid VAE-GANs (Dumoulin et al., 2016; Makhzani et al., 2016; Rosca et al., 2019; Zhao et al., 2017b) maximizes the log-likelihood of the data ($x \sim p(x)$) akin to VAEs. The log-likelihood, assuming the latent space to be distributed according to $p(z)$,

$$\log(p_\theta(x)) = \log\left(\int p_\theta(x|z)p(z)dz\right). \tag{1}$$

Here, $p(z)$ is usually Gaussian. This requires the generator $G_\theta$ to generate samples that assign high likelihood to every example x in the data distribution for a likely $z \sim p(z)$. Thus, the decoder $\theta$ can be forced to cover all modes of the data distribution $x \sim p(x)$. In contrast, GANs never directly maximize the data likelihood and there is no direct incentive to cover all modes.

However, the integral in (1) is intractable. VAEs and Hybrid VAE-GANs use amortized variational inference using a recognition network $q_\phi(z|x)$ ($R_\phi$). The final hybrid VAE-GAN objective of the state-of-the-art $\alpha$-GAN (Rosca et al., 2019) which integrates a synthetic likelihood ratio term is,

$$\mathcal{L}_{\alpha\text{-GAN}} = \lambda \, \mathbb{E}_{q_\phi(z|x)} \log(p_\theta(x|z)) \, + \, \mathbb{E}_{q_\phi(z|x)} \log\left(\frac{D_I(x|z)}{1 - D_I(x|z)}\right) - \text{KL}(p(z) \parallel q_\phi(z|x)). \tag{2}$$

This objective has two important shortcomings. Firstly, as pointed in (Bhattacharyya et al., 2018; Tolstikhin et al., 2018), this objective severely constrains the recognition network as the average likelihood of the samples generated from the posterior $q_\phi(z|x)$ is maximized. This forces all samples from $q_\phi(z|x)$ to explain x equally well, penalizing any variance in $q_\phi(z|x)$ and thus forcing it away from the Gaussian prior $p(z)$. Therefore, this makes it difficult to match the prior in the latent space and the encoder is forced to trade-off between a good estimate of the data log-likelihood and the divergence to the latent prior.

Secondly, the synthetic likelihood ratio term is the ratio of the output of $D_I$, any instability (non-smoothness) in the output of the classifier is magnified. Moreover, there is no incentive for $D_I$ to be smooth (stable). For two similar images, $\{x_1, x_2\}$ with $|x_1 - x_2| \le \epsilon$, the change of output $|D_I(x_1|z_1) - D_I(x_1|z_2)|$ can be arbitrarily large. This means that a small change in the generator output (e.g. after a gradient descent step) can have a large change in the discriminator output.

Next, we describe how we can effectively leverage multiple samples from $q_\phi(z|x)$ to deal with the first issue. Finally, we derive a stable synthetic likelihood term (Rosca et al., 2019; Wood, 2010) to deal with the second issue.

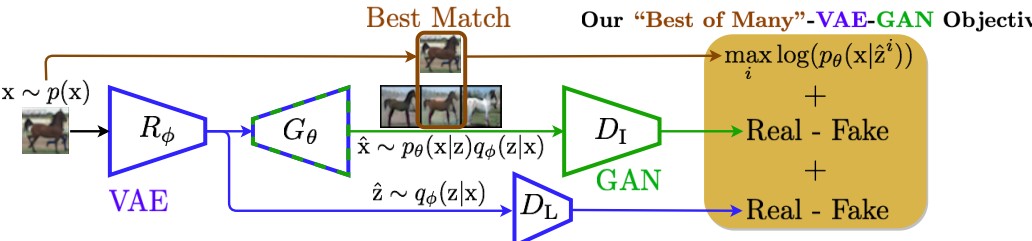

Figure 1: Overview of our BMS-VAE-GAN framework. The terms of our novel objective (7) are highlighted at the right. We consider only the best sample from the generator $G_\theta$ while computing the reconstruction loss.

## 3.2 LEVERAGING MULTIPLE SAMPLES

Building upon Bhattacharyya et al. (2018), we derive an alternative variational approximation of (1), which uses multiple samples to relax the constrains on the recognition network (full derivation in Appendix A),

$$\mathcal{L}_{\text{MS}} = \log \Big( \int p_\theta(\mathbf{x}|\mathbf{z}) q_\phi(\mathbf{z}|\mathbf{x}) \, dz \Big) - \text{KL}(q_\phi(\mathbf{z}|\mathbf{x}) \parallel p(z)). \tag{3}$$

In comparison to the $\alpha$-GAN objective (2) where the expected likelihood assigned by each sample to the data point x was considered, we see that in (3) the likelihood is computed considering all generated samples. The recognition network gets multiple chances to draw samples which assign high likelihood to x. This allows $q_\phi(\mathbf{z}|\mathbf{x})$ to have higher variance, helping it better match the prior and significantly reducing the trade-off with the data log-likelihood. Next, we describe how we can integrate a synthetic likelihood term in (3) to help us generate sharper images.

## 3.3 INTEGRATING STABLE SYNTHETIC LIKELIHOOD WITH THE "BEST-OF-MANY" SAMPLES

Considering only $L_1/L_2$ reconstruction based likelihoods $p_\theta(\mathbf{x}|\mathbf{z})$ (as in Bhattacharyya et al. (2018); Kingma & Welling (2014); Tolstikhin et al. (2018)) might not be sufficient in case of complex high dimensional distributions e.g. in case of image data this leads to blurry samples. Synthetic estimates of the likelihood Wood (2010) leverages a neural network (usually a classifer) which is jointly trained to distinguish between real and generated samples. The network is traiend to assign low likelihood to generated samples and higher likelihood to real data samples. Starting from (3), we integrate a synthetic likelihood term with weight $\beta$ to encourage our generator to generate realistic samples. The $L_1/L_2$ reconstruction likelihood (with weight $\alpha$) forces the coverage of all modes. However, unlike prior work (Bhattacharyya et al., 2019; Rosca et al., 2019), our synthetic likelihood estimator $D_I$ is not a classifier. We first convert the likelihood term to a likelihood ratio form which allows for synthetic estimates,

$$\begin{aligned} \mathcal{L}_{\text{MS}} =& \alpha \log \Big( \mathbb{E}_{q_\phi(\mathbf{z}|\mathbf{x})} p_\theta(\mathbf{x}|\mathbf{z}) \Big) + \beta \log \Big( \mathbb{E}_{q_\phi(\mathbf{z}|\mathbf{x})} p_\theta(\mathbf{x}|\mathbf{z}) \Big) - \text{KL}(q_\phi(\mathbf{z}|\mathbf{x}) \parallel p(z)) \\ \propto& \; \alpha \log \Big( \mathbb{E}_{q_\phi(\mathbf{z}|\mathbf{x})} \frac{p_\theta(\mathbf{x}|\mathbf{z})}{p(\mathbf{x})} \Big) + \beta \log \Big( \mathbb{E}_{q_\phi(\mathbf{z}|\mathbf{x})} p_\theta(\mathbf{x}|\mathbf{z}) \Big) - \text{KL}(q_\phi(\mathbf{z}|\mathbf{x}) \parallel p(z)). \end{aligned} \tag{4}$$

To enable the estimation of the likelihood ratio $p_\theta(\mathbf{x}|\mathbf{z})/p(\mathbf{x})$ using a neural network, we introduce the auxiliary variable y where, $y = 1$ denotes that the sample was generated and $y = 0$ denotes that the sample is from the true distribution. We can now express (4) (using Bayes theorem, see Appendix A),

$$\begin{aligned} =& \alpha \log \Big( \mathbb{E}_{q_\phi(\mathbf{z}|\mathbf{x})} \frac{p_\theta(\mathbf{x}|\mathbf{z}, y = 1)}{p(\mathbf{x}|y = 0)} \Big) + \beta \log \Big( \mathbb{E}_{q_\phi(\mathbf{z}|\mathbf{x})} p_\theta(\mathbf{x}|\mathbf{z}) \Big) - \text{KL}(q_\phi(\mathbf{z}|\mathbf{x}) \parallel p(z)). \\ =& \alpha \log \Big( \mathbb{E}_{q_\phi(\mathbf{z}|\mathbf{x})} \frac{p_\theta(y = 1|\mathbf{z}, \mathbf{x})}{1 - p(y = 1|\mathbf{x})} \Big) + \beta \log \Big( \mathbb{E}_{q_\phi(\mathbf{z}|\mathbf{x})} p_\theta(\mathbf{x}|\mathbf{z}) \Big) - \text{KL}(q_\phi(\mathbf{z}|\mathbf{x}) \parallel p(z)). \end{aligned} \tag{5}$$

The ratio $p_\theta(y=1|\mathbf{z},\mathbf{x})/1-p(y=1|\mathbf{x})$ should be high for generated samples which are indistinguishable from real samples and low otherwise. In case of image distributions, we find that direct estimation of

the numerator/denominator (as in Rosca et al. (2019)) exacerbates instabilities (non-smoothness) of the estimate. Therefore, we estimate this ratio directly using the neural network $D_I(x)$ – trained to produce high values for images indistinguishable from real images and low otherwise,

$$\mathcal{L}_{\text{MS-S}} \propto \alpha \log \left( \mathbb{E}_{q_\phi(z|x)} D_I(x|z) \right) + \beta \log \left( \mathbb{E}_{q_\phi(z|x)} p_\theta(x|z) \right) - \text{KL}(q_\phi(z|x) \parallel p(z)). \qquad (6)$$

To further unsure smoothness, we directly control the Lipschitz constant $K$ of $D_I$. This ensures, $\forall x_1, x_2, |D_I(x_1|z_1) - D_I(x_2|z_2)| \leq K|x_1 - x_2|$ – the function is strictly smooth everywhere. Small changes in generator output cannot arbitrarily change the synthetic likelihood estimate, hence allowing the generator to smoothly improve sample quality. We constrain the Lipschitz constant $K$ to 1 using Spectral Normalization Miyato et al. (2018). Note that the likelihood $p_\theta(x|z)$ takes the form $e^{-\lambda \|x-\hat{x}\|_n}$ in (6) – a log-sum-exp which is numerically unstable. As we perform stochastic gradient descent, we can deal with this after stochastic (MC) sampling of the data points. We can well estimate the log-sum-exp using the max – the "Best-of-Many-Samples" (Nielsen & Sun, 2016),

$$\log \left( \frac{1}{T} \sum_{i=1}^{i=T} p_\theta(x|\hat{z}^i) \right) \geq \max_i \log(p_\theta(x|\hat{z}^i)) - \log(T)$$

In practice, we observe that we can improve sharpness of generated images by penalizing generator $G_\theta$, using the least realistic of the $T$ samples,

$$\log \left( \sum_{i=1}^{i=T} D_I(x|\hat{z}^i) \right) \geq \min_i \log \left( D_I(x|\hat{z}^i) \right)$$

Our final "Best-of-Many"-VAE-GAN objective takes the form (ignoring the constant $\log(T)$ term),

$$\mathcal{L}_{\text{BMS-S}} = \alpha \min_i \log \left( D_I(x|\hat{z}^i) \right) + \beta \max_i \log(p_\theta(x|\hat{z}^i)) - \text{KL}(q_\phi(z|x) \parallel p(z)). \qquad (7)$$

We use the same optimization scheme as in Rosca et al. (2019). We provide the algorithm in detail in Appendix B.

**Approximation Errors.** The "Best-of-Many-Samples" scheme introduces the $\log(T)$ error term. However, this error term is dominated by the low data likelihood term in the beginning of optimization (Bhattacharyya et al., 2018). Later, as generated samples become more diverse, the log likelihood term is dominated by the Best of T samples – "Best of Many-Samples" is equivalent.

**Classifier based estimate of the prior term.** Recent work (Makhzani et al., 2016; Arjovsky et al., 2017; Rosca et al., 2019) has shown that point-wise minimization of the KL-divergence using its analytical form leads to degradation in image quality. Instead, KL-divergence term is recast in a synthetic likelihood ratio form minimized "globally" using a classifier instead of point-wise. Therefore, unlike Bhattacharyya et al. (2018), here we employ a classifier based estimate of the KL-divergence to the prior. However, as pointed out by prior work on hybrid VAE-GANs (Rosca et al., 2019), a classifier based estimate with still leads to mismatch to the prior as the trade-off with the data log-likelihood still persists without the use of the "Best-of-Many-Samples". Therefore, as we shall demonstrate next, the benefits of using the "Best-of-Many-Samples" extends to case when a classifier based estimate of the KL-divergence is employed.

## 4 EXPERIMENTS

Next, we evaluate on multi-modal synthetic data as well as CIFAR-10 and CelebA. We perform all experiments on a single Nvidia V100 GPU with 16GB memory. We use as many samples during training as would fit in GPU memory so that we make the same number of forward/backward passes as other approaches and minimize the computational overhead of sampling multiple samples.

Table 1: Evaluation on multi-modal synthetic data.

Table 2: Visualization of samples.

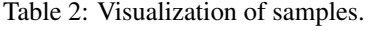

| | 2D Grid (25 modes) | | 2D Ring (8 modes) | |
|---|---|---|---|---|
| Method | Modes | HQ% | Modes | HQ% |
| VEEGAN (Srivastava et al., 2017) | 24.6 | 40 | 8 | 52.9 |
| GDPP-GAN (Elfeki et al., 2019) | 24.8 | 68.5 | 8 | 71.7 |
| SN-GAN (Miyato et al., 2018) | 23.8±1.5 | 90.9±4.0 | 6.8±1.1 | 86.6±9.7 |
| MD-GAN (Eghbal-zadeh et al., 2019) | 25 | 99.3±2.2 | 8 | 89.0±3.6 |
| WAE (Arjovsky et al., 2017) | 25 | 65.4±3.8 | 8 | 35.8±4.7 |
| $\alpha$-GAN (Rosca et al., 2019) | 25 | 70.5±4.2 | 8 | 83.6±5.3 |
| BMS-VAE-GAN (Ours) $T = 10$ | 25 | **99.7±0.2** | 8 | **99.6±0.3** |

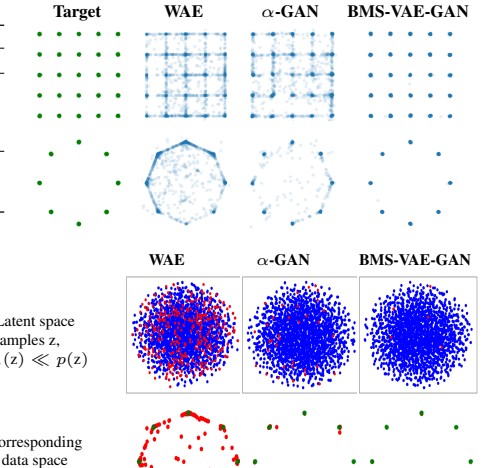

Table 3: Effect of our novel objective in the latent space. **Top Row:** The standard WAE and $\alpha$-GAN objectives leads to mismatch to the prior in the latent space. We show samples z (in red) which are highly likely under the standard Gaussian prior (blue) but have low probability under the learnt marginal posterior $q_\phi(z)$. **Bottom Row:** We show that such points z lead to low quality data samples (in red), which do correspond to any of the modes.

## 4.1 EVALUATION ON MULTI-MODAL SYNTHETIC DATA.

We evaluate in Tables 1 and 2 on the standard 2D Grid and Ring datasets, which are highly challenging due to their multi-modality. The metrics considered are the number of modes captured and % of high quality samples (within 3 standard deviations of a mode). The generator/discriminator architecture is same as in Srivastava et al. (2017). We see that our BMS-VAE-GAN (using the best of $T = 10$ samples) outperforms state of the art GANs e.g. (Eghbal-zadeh et al., 2019) and the WAE and $\alpha$-GAN baselines. The explicit maximization of the data log-likelihood enables our BMS-VAE-GAN and the WAE and $\alpha$-GAN baselines to capture all modes in both the grid and ring datasets. The significantly increased proportion of high quality samples with respect to WAE and $\alpha$-GAN baselines is due to our novel "Best-of-Many-Samples" objective. We illustrate this in Table 3. Following Rosca et al. (2019) we analyze the learnt latent spaces in detail, in particular we check for points (in red) which are likely under the Gaussian prior $p(z)$ (blue) but have low probability under the marginal posterior $q_\phi(z) = \int q_\phi(z|x)dx$. We use tSNE to project points from our 32-dimensional latent space to 2D. In Table 3 (Top Row) we clearly see that there are many such points in case of the WAE and $\alpha$-GAN baselines (note that this low probability threshold is common across all methods). In Table 3 (Bottom Row) we see that these points lead to the generation of low quality samples (in red) in the data space. Therefore, we see that our "Best-of-Many-Samples" samples objective helps us match the prior in the latent space and thus this leads to the generation of high quality samples and outperforming both state of the art GANs and hybrid VAE-GAN baselines.

Table 4: IvOM on Cifar10.

Table 5: Closest generated images found using IvOM.

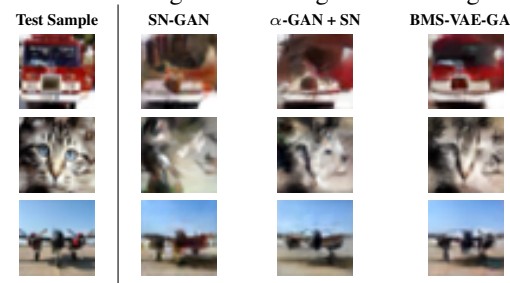

| Method | IvOM ↓ |
|---|---|
| DCGAN (Radford et al., 2016) | 0.0084±0.0020 |
| VEEGAN (Srivastava et al., 2017) | 0.0068±0.0001 |
| SN-GAN (Miyato et al., 2018) | 0.0055±0.0006 |
| $\alpha$-GAN + SN (Ours) $T = 1$ | 0.0048±0.0005 |
| BMS-VAE-GAN (Ours) $T = 30$ | **0.0037±0.0005** |

## 4.2 EVALUATION ON CIFAR-10

Next, we evaluate on the CIFAR-10 dataset. Auto-encoding based approaches (Kingma & Welling, 2014; Makhzani et al., 2016) do not perform well on this dataset, as a simple Gaussian reconstruction based likelihood is insufficient for such highly multi-modal image data. This makes CIFAR-10 a very challenging dataset for hybrid VAE-GANs.

**Architecture Details.** We use two different types of architectures for the generator/discriminator pair $G_\theta, D_\mathrm{I}$ : DCGAN based (Radford et al., 2016) as used in Rosca et al. (2019) and the Standard CNN used in Miyato et al. (2018); Tran et al. (2018).

**Experimental Details and Baselines.** We use the ADAM optimizer (Kingma & Ba, 2015) and use learning rate of $2 \times 10^{-4}$, $\beta_1 = 0.0$ and $\beta_2 = 0.9$ for all components. We use the same architecture of the latent space discriminator $D_\mathrm{L}$ as in $\alpha$-GAN Rosca et al. (2019) (3-layer MLP with 750 neurons each). Values of $\log(D_\mathrm{I}) \in [0, 2]$ work well.

We consider the following baselines for comparison against our BMS-VAE-GAN with a DCGAN generator/discriminator, 1. A standard DCGAN (Radford et al., 2016), 2. The $\alpha$-GAN model of (Rosca et al., 2019). Furthermore, we compare our BMS-GAN with the Standard CNN generator/discriminator to, 1. SN-GAN (Miyato et al., 2018), 2. BW-GAN (Adler & Lunz, 2018), 3. Dist-GAN (Tran et al., 2018), 4. Our $\alpha$-GAN + SN is an improved version of the $\alpha$-GAN which includes Spectral Normalization for stable estimation of synthetic likelihoods. Again, the $\alpha$-GAN and $\alpha$-GAN + SN baselines are identical to the corresponding BMS-VAE-GAN except for the "Best-of-Many-Samples" reconstruction likelihood.

**Discussion of Results.** We report results in Table 6. Please note that the higher latent space dimensionality (100) makes the latent spaces much harder to reliably analyze. Therefore, we rely on the FID and IoVM metrics. We follow evaluation protocol of Miyato et al. (2018); Tran et al. (2018) and use 10k/5k real/generated samples to compute the FID score. The $\alpha$-GAN (Rosca et al., 2019) model with (DCGAN architecture) demonstrates better fit to the true data distribution (29.3 vs 30.7 FID) compared to a plain DCGAN. This again shows the ability of hybrid VAE-GANs in improving the performance of plain GANs. We observe that our novel "Best-of-Many-Samples" optimization scheme outperforms both the plain DCGAN and hybrid $\alpha$-GAN(28.8 vs 29.4 FID), confirming the advantage of using "Best-of-Many-Samples". Furthermore, we see that our BMS-VAE outperforms the state-of-the-art plain auto-encoding WAE (Tolstikhin et al., 2018).

| Method | FID $\downarrow$ |
|---|---|
| DCGAN Architecture | |
| WAE (Tolstikhin et al., 2018) | 89.3±0.3 |
| BMS-VAE (Ours) $T = 10$ | 87.9±0.4 |
| DCGAN (Radford et al., 2016) | 30.7±0.2 |
| $\alpha$-GAN (Rosca et al., 2019) | 29.4±0.3 |
| BMS-GAN (ours) $T = 10$ | **28.8±0.4** |
| Standard CNN Architecture | |
| SN-GAN (Miyato et al., 2018) | 25.5 |
| BW-GAN (Adler & Lunz, 2018) | 25.1 |
| $\alpha$-GAN + SN (Ours) $T = 1$ | 24.6±0.3 |
| BMS-VAE-GAN (Ours) $T = 10$ | 23.8±0.2 |
| BMS-VAE-GAN (Ours) $T = 30$ | **23.4±0.2** |
| Dist-GAN (Tran et al., 2018) | 22.9 |
| BMS-VAE-GAN (Ours) $T = 10$ | **21.8±0.2** |

Table 6: FID on CIFAR-10.

We further compare our BMS-VAE-GAN to state-of-the-art GANs using the Standard CNN architecture in Table 6 with 100k generator iterations. Our $\alpha$-GAN + SN ablation significantly outperforms the state-of-the-art plain GANs (Adler & Lunz, 2018; Miyato et al., 2018), showing the effectiveness of hybrid VAE-GANs with a stable direct estimate of the synthetic likelihood on this highly diverse dataset. Furthermore, our BMS-VAE-GAN model trained using the best of $T = 30$ samples significantly improves over the $\alpha$-GAN + SN baseline (23.4 vs 24.6 FID), showing the effectiveness of our "Best-of-Many-Samples". We also compare to Tran et al. (2018) using 300k generator iterations, again outperforming by a significant margin (21.8 vs 22.9 FID). The IoVM metric of Srivastava et al. (2017) (Tables 4 and 5), illustrates that we are also able to better reconstruct the image distribution. The improvement in both sample quality as measured by the FID metric and data reconstruction as measured by the IoVM metric shows that our novel "Best-of-Many-Samples" objective helps us both match the prior in the latent space and achieve high data log-likelihood at the same time.

## 4.3 EVALUATION ON CELEBA

Next, we evaluate on CelebA at resolutions 64×64 and 128×128.

**Training and Architecture Details.** As the focus is to evaluate objectives for hyrid VAE-GANs, we use simple DCGAN based generators and discriminators for generation at both 64×64 and 128×128. Approaches like progressive growing (Karras et al., 2018) are orthogonal and can be applied on top.

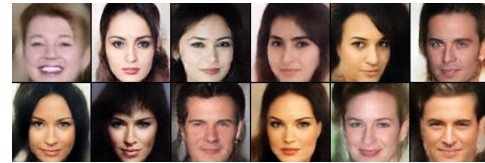

(a) Our $\alpha$-GAN + SN ($T = 1$, 128×128)      (b) Our BMS-VAE-GAN ($T = 10$, 128×128)

Figure 2: CelebA Random Samples. Our "Best of Many" reconstruction cost leads to sharper results.

**Baselines and Experimental Details.** We consider the following baselines for comparison with our BMS-GAN with $T = \{10, 30\}$ samples, 1. WAE (Tolstikhin et al., 2018) the state-of-the-art plain auto-encoding generative model, 2. $\alpha$-GAN (Rosca et al., 2019) the state-of-the-art hybrid VAE-GAN, 3. Our $\alpha$-GAN + SN is an improved version of the $\alpha$-GAN which includes Spectral Normalization for stable estimation of synthetic likelihoods. Note, the $\alpha$-GAN baseline is identical to our BMS-GAN except for the "Best-of-Many" reconstruction likelihood. Moreover, we include several plain GAN baselines, 1. Wasserstein GAN with gradient penalty (WGAN-GP) Gulrajani et al. (2017), 2. Spectral Normalization GAN (SN-GAN) Miyato et al. (2018), 3. Dist-GAN (Tran et al., 2018).

To train our BMS-VAE-GAN and $\alpha$R-GAN models we use the two time-scale update rule (Heusel et al., 2017) with learning rate of $1 \times 10^{-4}$ for the generator and $4 \times 10^{-4}$ for the discriminator. We use the Adam optimizer with $\beta_1 = 0.0$ and $\beta_2 = 0.9$. We use a three layer MLP with 750 neurons as the latent space discriminator $D_L$ (as in Rosca et al. (2019)) and a DCGAN based recognition network $R_\phi$. We use the hinge loss to train $D_I$ to produce high values for real images and low values for generated images, $\log(D_I) \in [-0.5, 0.5]$ works well.

**Discussion of Results.** We train all models for 200k iterations and report the FID scores (Heusel et al., 2017) for all models using 10k/10k real/generated samples in Table 7. The pure auto-encoding based WAE (Tolstikhin et al., 2018) has the weakest performance due to blurriness. Our pure auto-encoding BMS-VAE (without synthetic likelihoods) improves upon the WAE (39.8 vs 41.2 FID), already demonstrating the effectiveness of using "Best-of-Many-Samples". We see that the base DCGAN has the weakest performance among the GANs. BEGAN suffers from partial mode collapse. The SN-GAN improves upon WGAN-GP, showing the effectiveness of Spectral Normalization. However, there exists considerable artifacts in its generations. The $\alpha$-GAN of Rosca et al. (2019), which integrates the base DCGAN in its framework performs significantly better (31.1 vs 19.2 FID). This shows the effectiveness of VAE-GAN frameworks in increasing quality and diversity of generations. Our enhanced $\alpha$-GAN + SN regu-

| Method | FID ↓ |
|---|---|
| Resolution: 64×64 | |
| WAE (Tolstikhin et al., 2018) | 41.2±0.3 |
| BMS-VAE (Ours) $T = 10$ | 39.8±0.3 |
| DCGAN | 31.1±0.9 |
| WGAN-GP (Gulrajani et al., 2017) | 26.8±1.2 |
| BEGAN (Berthelot et al., 2017) | 26.3±0.9 |
| Dist-GAN (Tran et al., 2018) | 23.7±0.3 |
| SN-GAN (Miyato et al., 2018) | 21.9±0.8 |
| $\alpha$-GAN (Rosca et al., 2019) | 19.2±0.8 |
| $\alpha$-GAN + SN (Ours) $T = 1$ | 15.1±0.2 |
| BMS-VAE-GAN (Ours) $T = 10$ | 14.3±0.4 |
| BMS-VAE-GAN (Ours) $T = 30$ | **13.6±0.4** |
| Resolution: 128×128 | |
| SN-GAN (Miyato et al., 2018) | 60.5±1.5 |
| $\alpha$R-GAN (Ours) $T = 1$ | 45.8±1.4 |
| BMS-GAN (Ours) $T = 10$ | **42.7±1.2** |

Table 7: FID on CelebA.

larized with Spectral Normalization performs significantly better (15.1 vs 19.2 FID). This shows the effectiveness of a regularized direct estimate of the synthetic likelihood. Using the gradient penalty regularizer of Gulrajani et al. (2017) lead to drop of 0.4 FID. Our BMS-VAE-GAN improves significantly over the $\alpha$-GAN + SN baseline using the "Best-of-Many-Samples" (13.6 vs 15.1 FID). The results at 128×128 resolution mirror the results at 64×64. We additionally evaluate using the IoVM metric in Appendix C. We see that by using the "Best-of-Many-Samples" we obtain sharper (Figure 4d) results that cover more of the data distribution as shown by both the FID and IoVM.

## 5    CONCLUSION

We propose a new objective for training hybrid VAE-GAN frameworks which overcomes key limitations of current hybrid VAE-GANs. We integrate, 1. A "Best-of-Many-Samples" reconstruction likelihood which helps in covering all the modes of the data distribution while maintaining a latent space as close to Gaussian as possible, 2. A stable estimate of the synthetic likelihood ratio.. Our hybrid VAE-GAN framework outperforms state-of-the-art hybrid VAE-GANs and plain GANs in generative modelling on CelebA and CIFAR-10, demonstrating the effectiveness of our approach.

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

## APPENDIX A. ADDITIONAL DERIVATIONS

We begin with a derivation of the multi-sample objective (3). We maximize the log-likelihood of the data ($x \sim p(x)$). The log-likelihood, assuming the latent space to be distributed according to $p(z)$,

$$\log(p_\theta(x)) = \log\left(\int p_\theta(x|z)p(z)dz\right). \tag{8}$$

Here, $p(z)$ is usually Gaussian. However, the integral in (8) is intractable. VAEs and Hybrid VAE-GANs use amortized variational inference using an (approximate) variational distribution $q_\phi(z|x)$ (jointly learned),

$$\log(p_\theta(x)) = \log\left(\int p_\theta(x|z)\frac{p(z)}{q_\phi(z|x)}q_\phi(z|x)dz\right).$$

To arrive at a tractable objective, the standard VAE objective applies the Jensen inequality at this stage, but this forces the final objective to consider the average data-likelihood. Following Bhattacharyya et al. (2018), we apply the Mean Value theorem of Integration (Comenetz, 2002) to leverage multiple samples,

$$\log(p_\theta(x)) \geq \log\left(\int_a^b p_\theta(x|z)\, q_\phi(z|x)\, dz\right) + \log\left(\frac{p(z')}{q_\phi(z'|x)}\right),\ z' \in [a,b]. \tag{9}$$

We can lower bound (9) with the minimum value of $z'$,

$$\log(p_\theta(x)) \geq \log\left(\int_a^b p_\theta(x|z)\, q_\phi(z|x)\, dz\right) + \min_{z' \in [a,b]} \log\left(\frac{p(z')}{q_\phi(z'|x)}\right). \tag{S2}$$

As the term on the right is difficult to estimate, we approximate it using the KL divergence (as in Bhattacharyya et al. (2018)). Intuitively, as the KL divergence heavily penalizes $q_\phi(z|x)$ if it is high for low values $p(z)$, this ensures that the ratio $p(z')/q_\phi(z'|x)$ is maximized. Similar to Bhattacharyya et al. (2018), this leads to the "many-sample" objective (4) of the main paper,

$$\mathcal{L}_{\mathrm{MS}} = \log\left(\mathbb{E}_{q_\phi(z|x)}p_\theta(x|z)\right) - \mathrm{KL}(q_\phi(z|x)\ \|\ p(z)). \tag{4}$$

Next, we provide a detailed derivation of (5). Again, to enable the estimation of the likelihood ratio $p_\theta(x|z)/p(x)$ using a neural network, we introduce the auxiliary variable y where, $y = 1$ denotes that the sample was generated and $y = 0$ denotes that the sample is from the true distribution. We can now express (5) as (using Bayes theorem),

$$\alpha \log\left(\mathbb{E}_{q_\phi(z|x)}\frac{p_\theta(x|z, y = 1)}{p(x|y = 0)}\right) + \beta \log\left(\mathbb{E}_{q_\phi(z|x)}p_\theta(x|z)\right) - \mathrm{KL}(q_\phi(z|x)\ \|\ p(z)).$$

$$= \alpha \log\left(\mathbb{E}_{q_\phi(z|x)}\frac{p_\theta(y = 1|z, x)}{1 - p(y = 1|x)}\right) + \beta \log\left(\mathbb{E}_{q_\phi(z|x)}p_\theta(x|z)\right) - \mathrm{KL}(q_\phi(z|x)\ \|\ p(z)).$$

This is because, (assuming independence $p(z, x) = p(z)p(x)$ )

$$p_\theta(x|z, y = 1) = \frac{p(y = 1|z, x)p(x)}{p(y = 1)}$$

and,

$$p_\theta(x|y = 0) = \frac{p(y = 0|x)p(x)}{p(y = 0)}.$$

Assuming, $p(y = 0) = p(y = 1)$ (equally likely to be true or generated),

$$\frac{p_\theta(x|z, y = 1)}{p(x|y = 0)} = \frac{p_\theta(y = 1|z, x)}{p(y = 0|x)}.$$

---

**Algorithm 1:** BMS-VAE-GAN Training.

---

1  Initialize parameters of $R_\phi, G_\theta, D_\mathrm{I}, D_\mathrm{L}$;
2  **for** $i \leftarrow 0$ **to** *max_iters* **do**
3     Update $R_\phi, G_\theta$ (jointly) using our $\mathcal{L}_{\text{BMS-S}}$ objective;
4     Update $D_\mathrm{I}$ using hinge loss to produce high values ($\geq a$) for real images and low ($\leq b$)
         otherwise: $\mathbb{E}_{p(\mathbf{x})} \max\{0, a - \log(D_\mathrm{I}(\mathbf{x}))\} + \mathbb{E}_{p(\mathbf{z})} \max\{0, b + \log(D_\mathrm{I}(G_\theta(\mathbf{z})))\}$;
5     Update $D_\mathrm{L}$ using the standard cross-entropy loss:
         $\mathbb{E}_{p(\mathbf{z})} \log(D_\mathrm{L}(\mathbf{z})) + \mathbb{E}_{p(\mathbf{x})} \log(1 - D_\mathrm{L}(R_\phi(\mathbf{x})))$;
6  **end**

---

## APPENDIX B. TRAINING ALGORITHM

We detail in algorithm 1, how the components $R_\phi, G_\theta, D_\mathrm{I}, D_\mathrm{L}$ of our BMS-VAE-GAN (see Figure Figure 1) are trained. We follow Rosca et al. (2019) in designing algorithm 1. However, unlike Rosca et al. (2019), we train $R_\phi, G_\theta$ jointly as we found it to be computationally cheaper without any loss of performance. Also unlike Rosca et al. (2019), we use the hinge loss to update $D_\mathrm{I}$ as it leads to improved stability (as discussed in the main paper).

## APPENDIX C. ADDITIONAL RESULTS USING THE IoVM METRIC

We additionally evaluate using the IoVM on CelebA in Table 8, using the base DCGAN architecture at $64{\times}64$ resolution. We observe that our BMS-VAE-GAN performs better. The improvement is smaller compared to CIFAR-10 because CelebA is less multi-modal compared to CIFAR-10. However, we still observe better overall sample quality from our BMS-VAE-GAN. This means that although difference in data reconstruction is smaller, our BMS-VAE-GAN enables better match the prior in the latent space. Finally, we provide additional examples of closest matches found using IoVM in Figure 3, illustrating regions of the data distribution captured by BMS-VAE-GAN but not captured by SN-GAN or $\alpha$-GAN + SN.

| Method | IoVM $\downarrow$ |
|---|---|
| SN-GAN (Miyato et al., 2018) | $0.0221{\pm}0.0003$ |
| $\alpha$-GAN + SN (Ours) $T = 1$ | $0.0036{\pm}0.0001$ |
| BMS-VAE-GAN (Ours) $T = 10$ | **$0.0034{\pm}0.0001$** |

Table 8: Evaluation on CelebA using the IoVM metric.

## APPENDIX D. ADDITIONAL QUALITATIVE EXAMPLES ON CELEBA AND CIFAR-10

In Figure 4, we compare qualitatively our BMS-VAE-GAN against other state-of-the-art GANs. We use the same settings as in the main paper and use the same DCGAN architecture across methods (as the aim is to evaluate training objectives). Again note that, approaches like Karras et al. (2018) use more larger generator/discriminator architectures and can be applied on top. We see that BEGAN (Berthelot et al., 2017) produces sharp images (with only a few very minuscule artifacts), but lack diversity – also reflected by the lower FID score in Table 2 of the main paper. In comparison, both SN-GAN (Miyato et al., 2018) and Dist-GAN (Tran et al., 2018) produce sharp and diverse images (again reflected by the FID score in Table 2 of the main paper) but also introduce artifacts. Dist-GAN (Tran et al., 2018) introduces relatively fewer artifacts in comparison to SN-GAN (Miyato et al., 2018). In comparison, our BMS-VAE-GAN strikes the best balance – generating sharp and diverse images with few if any artifacts (also again reflected by the FID scores in the main paper).

We also provide additional qualitative examples on CIFAR-10 in Figure 5, highlighting sharper images compared to $\alpha$-GAN +SN.

| Test Sample | SN-GAN | $\alpha$-GAN + SN | **BMS-VAE-GAN** | Test Sample | SN-GAN | $\alpha$-GAN + SN | **BMS-VAE-GAN** |

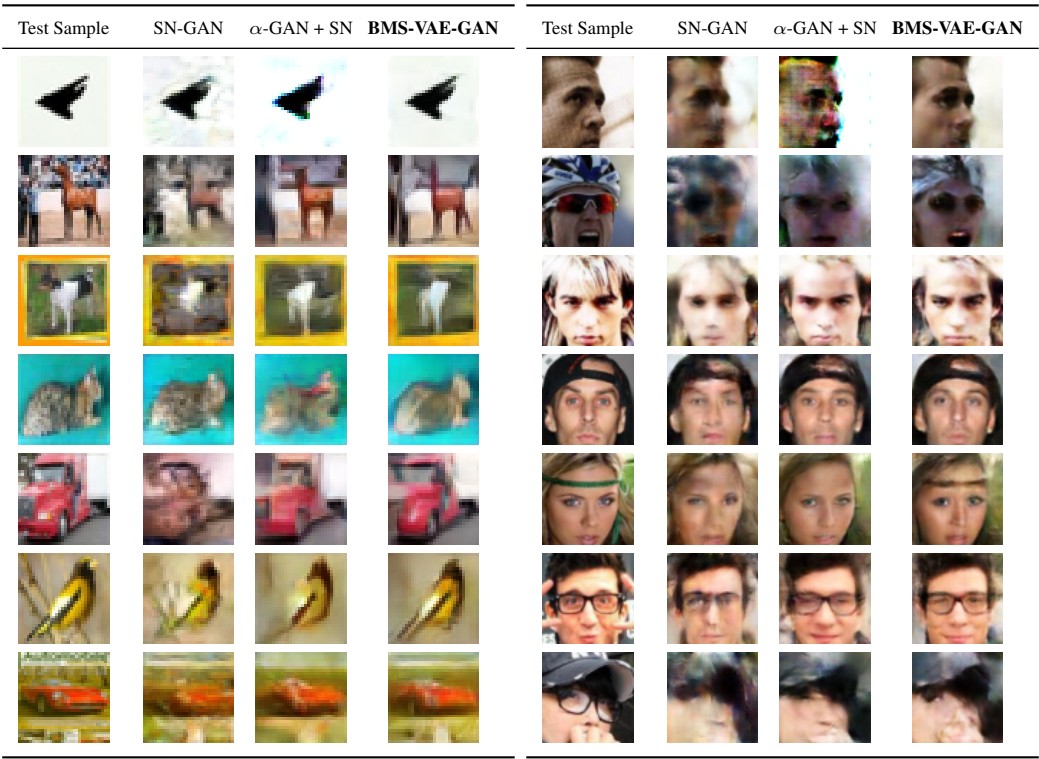

Figure 3: Closest images found by optimising in the latent space. Left: CIFAR-10, Right: CelebA.

## APPENDIX E. ADDITIONAL DIVERSITY EVALUATION USING LPIPS

In Table 9 we include diversity using the LPIPS metric. To compute the LPIPS diversity score 5k samples were randomly generated and the similarity within the batch was computed. We see that our BMS-VAE-GAN generates the most diverse examples on both datasets, further highlighting the effectiveness of our "Best-of-Many-Samples" objective.

| Method | CelebA $\downarrow$ | CIFAR-10 $\downarrow$ |
|---|---|---|
| SN-GAN | 0.160 | 0.148 |
| $\alpha$-GAN + SN (Ours) $T = 1$ | 0.162 | 0.145 |
| BMS-VAE-GAN (Ours) $T = 10$ | **0.151** | **0.140** |

Table 9: Evaluation using the LPIPS metric.

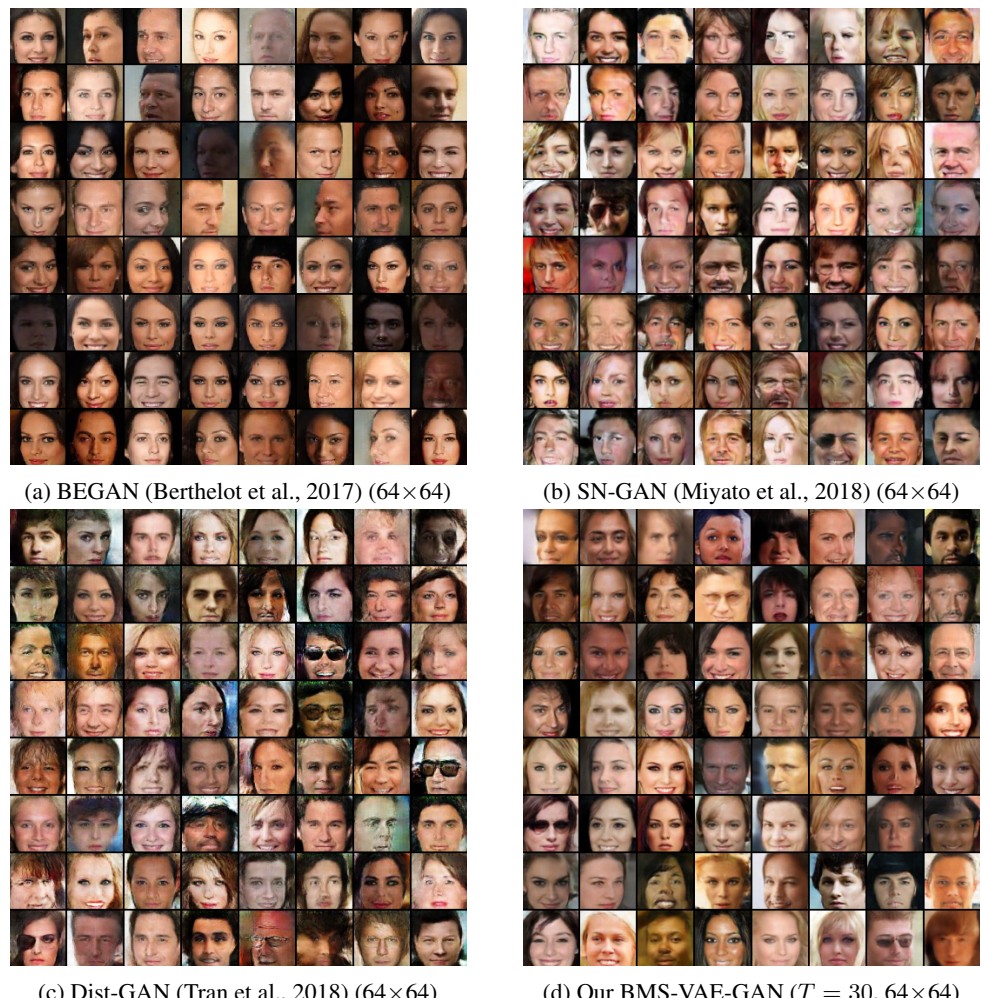

(a) BEGAN (Berthelot et al., 2017) (64×64)  (b) SN-GAN (Miyato et al., 2018) (64×64)

(c) Dist-GAN (Tran et al., 2018) (64×64)  (d) Our BMS-VAE-GAN ($T = 30$, 64×64)

Figure 4: CelebA Random Samples of state-of-the-art GANs versus our BMS-VAE-GAN (using DCGAN architecture).

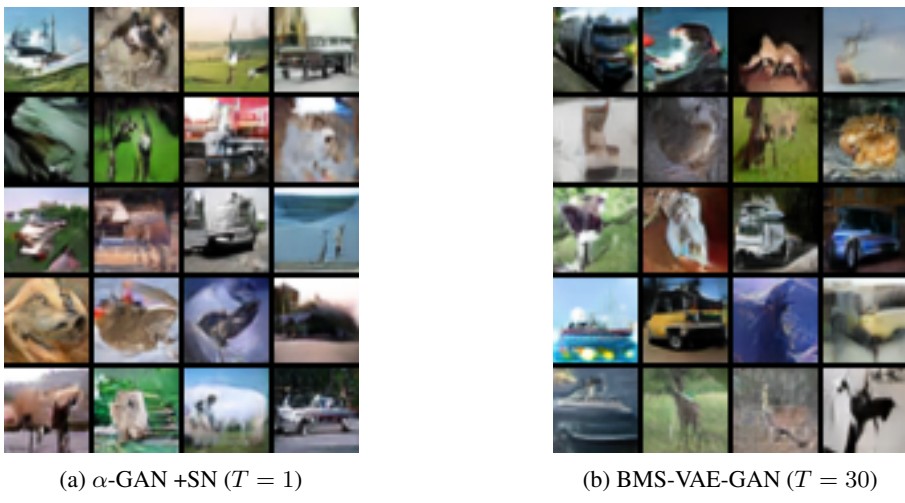

(a) $\alpha$-GAN +SN ($T = 1$)  (b) BMS-VAE-GAN ($T = 30$)

Figure 5: CIFAR-10 Random Samples (using Standard CNN architecture).

