# OpenReview forum: "``"Best-of-Many-Samples" Distribution Matching"
_ICLR.cc/2020/Conference — Reject_

### Official Review · AnonReviewer1 · 2019-10-21
**Official Blind Review #1**

**Rating:** 3

**Review:**

“Best of Many Samples” Distribution matching

Summary:

This paper proposes to a novel VAE-GAN hybrid which, during training, draws multiple samples from the reparameterized latent distribution for each inferred q(z|x), and only backpropagates reconstruction error for the resulting G(z) which has the lowest reconstruction. The authors appear to use the AAE method instead of analytically matching KL(p||q) for enforcing that the latents q(z|x) match the prior. The authors present results on MoG toy datasets, CIFAR-10, and CelebA, and compare  against several other models.

My take:

The idea in this paper is moderately interesting, well-founded, has plenty of precedent in the literature (while still being reasonably novel),  but the results present only a minimal improvement (a 5% relative improvement in FID over the baseline model from Rosca et al on CIFAR, especially when including SN [which is not a contribution of this paper]) and come at a substantial compute cost, requiring up to 30 extra samples per batch in order to attain this minimal increase. While I think the idea is interesting, the change in results over Rosca et. al does not seem to justify its increased computational expense (which is also not characterized in sufficient thoroughness). I am pretty borderline on this paper ( about a 5/10) but under the 1-3-6-8 scoring constraint I tend to lean reject because while I like the idea, I do not think the results are significant enough to support its adoption; I think the relative compute and implementation cost limit this method’s potential impact. I am keen to discuss this paper with the other reviewers.

Notes:

-The results on the 2D MoG toy datasets are good but are also suspect—the authors state that they use a 32-dimensional latent space, but the original code provided for VEEGAN uses a 2-dimensional latent space. The authors should re-run the experiment for BMS-VAE-GAN  using a 2D latent space (this should be very easy and take less than an hour on a GPU to get several runs in).

-“again outperforming by a significant margin (21.8 vs 22.9 FID)” This is not a significant margin; this is less than a 5% margin and, at those FID scores, represents an imperceptible change in sample quality.

-The authors seem to suggest that applying spectral norm to the GAN of Rosca et. al. is somehow a contribution (e.g. having “ours” next to this model in the tables); I would advise against even appearing to suggest this as it is clearly not a contribution.

-Characterize the increase in compute cost. “. We use as many samples during training as would fit in GPU memory so that we make the same number of forward/backward passes as other approaches and minimize the computational overhead of sampling multiple samples” is a qualitative description; I would like to see this quantitatively described. How do the runtimes differ between your baseline and the T=10 and T=30 runs? If they don’t differ, why? Are the authors e.g. reducing the batch size by a factor of 10 or 30 to make this computationally tractable?

-The latent space discriminator D_L should be referred to in section 3; its formal introduction is deferred to later in the paper, hampering the presentation and flow.

-CelebA is not multimodal; it is in fact, highly constrained, and primarily only has textural variation (virtually no pose variation).

-ALI and BiGAN are listed under Hybrid VAE-GANs. These models are not VAE-GAN hybrids. Additionally, this section states that BiGAN builds upon ALI. This is not true, these papers are in fact proposing the same thing and were released at nearly the exact same time.  Do not mischaracterize or incorrectly summarize papers. Please re-read both papers and refer to them correctly.

-Mode collapse (when many points in z map to an unexpectedly small region in G(z)) is a different phenomenon from mode dropping (when many points in x are not represented in G(z), i.e. no point in z maps to a cluster of x’s, as is the case if e.g. a celebA model generates frowning and neutral faces but no smiling faces). While these phenomena often co-occur (especially during complete training collapse), they are not the same thing, and this paper conflates them in several places.

Minor:

Section 3, paragraph 2: “The GAN (Gθ,DI…” There’s a close parenthesis missing here.

Section 3.3: “The network is traiend…”

Please thoroughly proofread your paper for typos and grammatical mistakes.



**Experience Assessment:**

I have published in this field for several years.

**Review Assessment: Checking Correctness Of Derivations And Theory:**

I carefully checked the derivations and theory.

**Review Assessment: Checking Correctness Of Experiments:**

I carefully checked the experiments.

**Review Assessment: Thoroughness In Paper Reading:**

I read the paper thoroughly.

---

### Official Review · AnonReviewer2 · 2019-10-21
**Official Blind Review #2**

**Rating:** 3

**Review:**

1.  Using Discriminator to estimate the likelihood ratio is a commonly used approach, which was first proposed in [1]. This is also generalized as a reversed KL based GAN in [2] [3]. The authors failed to discuss this with these previous works in Section 3.3 and in Related works.

2. How is the best of many comparing with importance sampling method? I think using importance sampling is the most intuitive baseline.

3. this paper is not well written. L_1/L_2 has never explained throughout this paper, also has typos such as "taiend".


[1] Adversarial Variational Bayes: Unifying Variational Autoencoders and Generative Adversarial Networks
[2] Variational Annealing of GANs: A Langevin Perspective
[3] Symmetric variational autoencoder and connections to adversarial learning

**Experience Assessment:**

I have published one or two papers in this area.

**Review Assessment: Checking Correctness Of Derivations And Theory:**

I assessed the sensibility of the derivations and theory.

**Review Assessment: Checking Correctness Of Experiments:**

I assessed the sensibility of the experiments.

**Review Assessment: Thoroughness In Paper Reading:**

I read the paper at least twice and used my best judgement in assessing the paper.

---

### Official Review · AnonReviewer3 · 2019-10-28
**Official Blind Review #3**

**Rating:** 6

**Review:**

This paper presents a new objective function for hybrid VAE-GANs. To overcome a number of known issues with VAE-GANs, this work uses multiple samples from the generator network to achieve a high data log-likelihood and low divergence to the latent prior.
In the experimental section, the ``"Best-of-Many-Samples" approach is shown to outperform other state-of-the-art methods on CIFAR-10 and a synthetic dataset.

Thanks for submitting code with your submission!

Caveat: I'm not an expert in this domain and did my best to review this paper.

Questions:
- Considering the smaller gap between α-GAN+SN and BMS-VAE-GAN, I was wondering how much of the improvement is due to spectral normalization vs using multiple samples. Did you do an ablation study of BMS-VAE-GAN without SN?
- I noticed some minor typos in the text. Please fix (3.2 "constrains" -> "constraints", 3.3 "traiend", 3.3 "unsure" -> "ensure").

**Experience Assessment:**

I do not know much about this area.

**Review Assessment: Checking Correctness Of Derivations And Theory:**

I did not assess the derivations or theory.

**Review Assessment: Checking Correctness Of Experiments:**

I did not assess the experiments.

**Review Assessment: Thoroughness In Paper Reading:**

I read the paper at least twice and used my best judgement in assessing the paper.

---

### Decision · Program_Chairs · 2019-12-19

**Decision:**

Reject

**Comment:**

This paper proposed an improvement on VAE-GAN which draws multiple samples from the reparameterized latent distribution for each inferred q(z|x), and only backpropagates reconstruction error for the resulting G(z) which has the lowest reconstruction.  While the idea is interesting, the novelty is not high compared with existing similar works, and the improvement is not significant.